# Emulating the Delivery of Sawtooth Proton Arc Therapy Plans on a Cyclotron-Based Proton Beam Therapy System

**DOI:** 10.3390/cancers16193315

**Published:** 2024-09-27

**Authors:** Samuel Burford-Eyre, Adam Aitkenhead, Jack D. Aylward, Nicholas T. Henthorn, Samuel P. Ingram, Ranald Mackay, Samuel Manger, Michael J. Merchant, Peter Sitch, John-William Warmenhoven, Robert B. Appleby

**Affiliations:** 1Department of Physics and Astronomy and the Cockcroft Institute, School of Natural Sciences, Faculty of Science and Engineering, The University of Manchester, Manchester M13 9PL, UK; 2Christie Medical Physics and Engineering, The Christie NHS Foundation Trust, Manchester M20 4BX, UK; 3Division of Cancer Sciences, Faculty of Biology Medicine and Heath, The University of Manchester, Manchester M13 9PL, UK; 4School of Applied Sciences, University of the West of England, Bristol BS16 1QY, UK

**Keywords:** proton arc therapy, intensity modulated proton therapy, delivery, energy layer switching, gantry motion, sawtooth arc

## Abstract

**Simple Summary:**

The overall delivery time of proton arc therapy (PAT) plans on current clinical systems must be evaluated due to high upward energy layer switching times (ELSTs) in order to identify clinically suitable methods of PAT planning and delivery. We present the application of an emulator for modelling the delivery of ‘sawtooth’ PAT plans on an existing cyclotron-based system. We show that this method of PAT planning consistently requires a longer delivery time than static intensity modulated proton therapy (IMPT) and that delivering PAT using a continuous gantry rotation remains the optimum delivery method on such systems. This analysis shows that the delivery of PAT plans generated using the simplified sawtooth PAT planning approach may be clinically infeasible without further developments to the existing clinical technologies.

**Abstract:**

**Purpose**: To evaluate and compare the deliverability of ‘sawtooth’ proton arc therapy (PAT) plans relative to static intensity modulated proton therapy (IMPT) at a cyclotron-based clinical facility. **Methods**: The delivery of single and dual arc Sawtooth PAT plans for an abdominal CT phantom and multiple clinical cases of brain, head and neck (H&N) and base of skull (BoS) targets was emulated under the step-and-shoot and continuous PAT delivery regimes and compared to that of a corresponding static IMPT plan. **Results**: Continuous PAT delivery increased the time associated with beam delivery and gantry movement in single/dual PAT plans by 4.86/7.34 min (brain), 7.51/12.40 min (BoS) and 6.59/10.57 min (H&N) on average relative to static IMPT. Step-and-shoot PAT increased this delivery time further by 4.79 min on average as the delivery was limited by gantry motion. **Conclusions**: The emulator can approximately model clinical sawtooth PAT delivery but requires experimental validation. No clear benefit was observed regarding beam-on time for sawtooth PAT relative to static IMPT.

## 1. Introduction

The recent developments in accelerator-related technologies enabling the clinical delivery of spot-scanned intensity modulated proton therapy (IMPT) [1,2,3,4] have increased the feasibility of applying arc-like delivery techniques to proton beam therapy (PBT). However, the dosimetric and delivery benefits of proton arc therapy (PAT) over the current clinical standard of fixed-field (static) IMPT are yet to be established [5,6,7,8,9,10,11,12,13,14,15,16,17,18]. This is primarily due to the high (4.5–30.0 s) upward energy layer switching times of the existing clinical systems creating a Pareto front between the dosimetric quality and overall delivery time of a PAT plan [18]. While a range of planning algorithms have been used in many dosimetric PAT planning studies [5,14,15,16,17,18,19], the deliverability of these plans on current PBT systems must be evaluated in order to assess the advantages and clinical suitability of each method of PAT planning and delivery.

PAT involves the delivery of a beam consisting of individual energy layers across a pre-defined angular sector around the patient. The PAT plan delivery can be categorised as ‘continuous’ or ‘step-and-shoot’, depending on whether beam delivery and gantry rotation occur simultaneously or not. During the delivery of each energy layer, the dose is delivered to the tumour via beam spots that are scanned across an array of pre-determined locations. Once delivery of an energy layer is complete, energy layer switching occurs by inserting/retracting graphite wedges into the beam and modifying magnet currents to appropriately narrow the energy spectrum, steer and focus the beam [20]. Due to magnetic hysteresis and distance the wedges are required to be inserted/retracted, this process requires a variable amount of time depending on the magnitude of the energy switch and initial energy [21]. Therefore, for a PAT delivery emulator to be able to model the beam delivery time, it must include a model of variable energy layer switching times (ELSTs), as well as gantry motion, spot switching and spot deliver times.

Additional parameters may also be required to increase a model’s specificity to a given clinical system. For example, models of the ProteusPLUS^®^ [22,23] and ProteusONE^®^ [21] systems (IBA: Louvain-La-Neuve, Belgium) have been developed and are able to predict beam delivery times (BDTs) to within a mean and standard deviation of −0.74 ± 3.33% and 2.1 ± 3.0%, respectively. To date, multiple studies involving many PAT treatment planning algorithms have used computational modelling to evaluate the overall treatment delivery time of PAT plans relative to static IMPT. The majority of these use various iterations of the ‘Spot-scanning proton arc therapy’ (SPArc) PAT planning algorithm. The initial SPArc plans, aiming to treat lung and oropharyngeal cancers, increased BDTs by between 59 to 591 s and 33 to 876 s for cases of lung and oropharyngeal cancer [5], using the assumption that ELSTs were constant and ranged from 0.5–4.0 s. However, after multiple improvements to the SPArc algorithm and the development of centre-specific delivery models, the delivery times of SPArc plans have been subsequently reduced by 16–26% [24] and have shown a 58% reduction relative to static IMPT [25]. Such BDT reductions may improve the patient treatment experience and increase patient throughput in single gantry PBT facilities. Other studies involving alternate PAT planning algorithms have shown that PAT increases BDTs by up to 214 s, 47 s, 54 s and 49 s for prostate, lung, brain and liver cancers, respectively [15,18]. As IMPT is commonly used to treat paediatric patients under anaesthesia [26], the risk of such BDT increases to patients should be carefully evaluated. Simulation studies modelling PAT delivery under a continuous gantry rotation were found to reduce PAT treatment times by 26–67% relative to step-and-shoot PAT delivery, across multiple treatment sites [5,6]. Therefore, in order to ensure the clinical feasibility of PAT during planning, the BDT should be directly evaluated or optimised by an appropriate treatment planning system.

Due to the inability of current clinical systems to continuously deliver PAT and the lack of PAT specific QC protocols, few studies aim to experimentally verify the delivery of existing PAT plans. Li et al. [27] developed the proton dynamic arc delivery (PDAD) module for an IBA ProteusONE^®^ system and delivered a brain SPArc plan in 4 min, showing 98.3% consistency (for gamma criteria of 1%, 1 mm) between the measured and planned dose distributions. A computational model of the effects of delivering PAT plans using the PDAD module was later developed [25]. Currently, all existing studies analysing the deliverability of PAT plans utilise specific parameters taken from single-room IBA Proteus^®^ systems. However, approximately 13% of the PBT facilities currently in operation use a multi-room ProBeam^®^ system (Varian: Palo Alto, CA, USA) [28], which has the flexibility to benefit more from the potential logistical improvements of PAT relative to static IMPT. The deliverability of PAT on such systems is yet to be investigated.

We present a new IMPT and PAT delivery emulator for the cyclotron-based Varian ProBeam^®^ PBT system [29], that enables a detailed comparison between the individual aspects of PAT delivery relative to static IMPT as well as an analysis of the dosimetric consequences of continuous PAT delivery. The use of this emulator is demonstrated for the delivery of the previously established static IMPT and sawtooth PAT plans on an abdominal CT phantom as well as two clinical cases of brain, base of skull (BoS) and head and neck (H&N) cancers [30].

## 2. Materials and Methods

A PAT emulator was developed using matRad v3.0.0 [31] to model the delivery of sawtooth PAT plans under the step-and-shoot and continuous delivery regimes on a clinical gantry. This supports the delivery of single, dual and partial arc PAT, as well as the use of multiple targets. A separate static IMPT emulator was also developed using the same approach to enable a comparison of the overall delivery times between PAT and the current clinical standard of static IMPT. Both emulators utilise experimental and Monte-Carlo [32,33] derived measurements of the ELST and MU delivery rate in both the PBT Stoller research room (RR) and a clinical gantry at the Christie NHS Foundation Trust. Details of these are provided in the Appendix A.

The emulators were specifically designed to be implemented after the PAT plans were generated using the sawtooth PAT planning algorithm [30]. This algorithm aims to apply the general methods of conventional IMPT treatment planning to PAT through the use of dose-influence matrices, robust optimisation, spot reduction and DVH/dose metric analysis. During energy layer selection, an initial set of energy layers is calculated from user-defined static field angles in order to promote adequate target coverage. These energy layers are then interpolated across the angular sectors between the static fields, creating a sawtooth-like pattern across the range of gantry angles utilised by the arc delivery.

After a Sawtooth PAT plan has been calculated, the weightings, angles and positions of each spot relative to the isocentre are used as inputs into the emulator. First, spot weightings are scaled in order to model the delivery of a single treatment fraction. Next, the gantry motion is calculated under the step-and-shoot and continuous delivery regimes. Step-and-shoot delivery requires the gantry to stop in order to deliver each energy layer, while continuous delivery allows the gantry to rotate during delivery, only stopping when an upward energy layer switch is required. Both regimes assume a constant maximum gantry acceleration and velocity of $\alpha_{\max}$ = 0.6°s^−2^ and $\omega_{\max}$ = 6°s^−1^, respectively, as per the parameters of the existing clinical gantries [6].

### 2.1. Mathematical Details of Sawtooth PAT

In the following section, we describe the details of IMPT and PAT emulation using the following notation:*F* is the set of fields or ‘teeth’ within the treatment plan for IMPT and PAT, respectively, in the order corresponding with delivery;*E* is the set of all energy layers contained within the plan;$E_{f}$ is the set of energy layers contained within field/tooth f∈F in descending order of energy;*S* is the set of all spots contained within the plan;$S_{e}$ is the set of all spots contained within layer e∈E.

The total time required to deliver spot *i* is separated into 3 components:(1)$$t_{\mathrm{spot},i}=t_{\mathrm{scan},i}+t_{\mathrm{del},i}+t_{\mathrm{switch},i},\;i\in S\;$$
where $t_{\mathrm{scan}}$, $t_{\mathrm{del}}$ and $t_{\mathrm{switch}}$ are the times required to scan the spot to the correct position, deliver the MUs associated with the spot and switch beam energy (if required) respectively. Here,
(2)$$t_{\mathrm{scan},i}=\frac{\sqrt{{\left(x_{i}-x_{i-1}\right)}^{2}+{\left(y_{i}-y_{i-1}\right)}^{2}}}{v_{\mathrm{scan}}},\;i\in S\;$$
where $x_{i}$ and $y_{i}$ are the spot coordinates, and $v_{\mathrm{scan}}$ is the spot scanning speed. As both $t_{\mathrm{del}}$ and $t_{\mathrm{switch}}$ are energy/system dependent, measurements were taken in the Christie NHS Foundation Trust’s RR and on a clinical gantry to determine their values and associated errors. Details of this process are provided in the Appendix A. The spot scanning speed in the RR and clinical gantries was determined to be 10 ms^−1^.

During static IMPT delivery emulation, the gantry angle is fixed during delivery such that the spot positions are consistent with the original plan. Between fields, the gantry is assumed to rotate at its maximum acceleration over the first half of the angular sector separating the fields, $\Delta \theta_{f}$, before decelerating over the remaining half such that its rotation speed is
(3)$$\omega =\sqrt{\Delta \theta_{f}\;\alpha_{\max}},\;f\in F,$$
under the constraint that ω cannot exceed $\omega_{\max}$. The additional dead-time where energy switching has finished but the gantry and nozzle are still moving to the position required to deliver the next field is then defined as
(4)$$t_{\mathrm{dead},f}=2\sqrt{\frac{\Delta \theta_{f}}{\alpha_{\max}}}+t_{\mathrm{noz},f}-t_{\mathrm{switch},f},\;f\in F,$$
where the nozzle moving time is
(5)$$t_{\mathrm{noz},f}=\frac{d_{\mathrm{retract}}-\mathrm{SAD}+{\mathrm{SSD}}_{f}-d_{\mathrm{clearance}}}{v_{\mathrm{noz}}},\;f\in F,$$
using the following definitions:$d_{\mathrm{retract}}$ = 42 cm: distance along beam axis between nozzle and isocentre when the nozzle is fully retracted;source-to-axis distance (SAD) = 243 cm: distance between the location of the ‘effective source’ (average steering magnet position) and isocentre along beam axis;source-to-surface distance (SSD_*f*_): distance along beam axis between the location of the MU effective source and patient surface at the angle corresponding to field *f*;$d_{\mathrm{clearance}}$ = 10 cm: the required minimum distance along the beam axis between the nozzle and the patient surface during delivery to avoid any risk of nozzle/patient collisions;$v_{\mathrm{noz}}$ = 1 cms^−1^: experimentally measured nozzle insertion/retraction speed.

$t_{\mathrm{dead},f}$ is set to 0 s if it is lower than $t_{\mathrm{switch},f}$, i.e. energy layer switching occurring simultaneously to gantry and nozzle motion. The overall delivery time of a static IMPT plan can then be approximated as
(6)$$T=\sum_{i\in S}t_{\mathrm{spot},i}+\sum_{f\in F}t_{\mathrm{dead},f}.$$

Similar to static IMPT, during step-and-shoot PAT delivery, each spot remains in the position assigned during planning, and the gantry motion follows Equation (Equation 3), where $\Delta \theta_{f}$ denotes the control point spacing between consecutive energy layers. However, the nozzle remains in a fixed position relative to the patient such that $d_{\mathrm{clearance}}$ = 10 cm at the angle where the patient surface is furthest from the isocentre along the delivery arc. Therefore, $t_{\mathrm{noz},f}$ = 0 such that the dead-time and overall delivery time for step-and-shoot PAT follow Equations (Equation 4) and (Equation 6), where each field *f* is equivalent to an individual energy layer *e*.

In contrast to step-and-shoot delivery, continuous PAT delivery causes spots to shift from their original (planned) locations. Therefore, the true angular positions of each spot must also be calculated prior to a final dose calculation. First, the total expected delivery time of each energy layer, before accounting for gantry motion, is calculated as
(7)$$t_{\mathrm{del},e}=\sum_{i\in Se}t_{\mathrm{spot},i}.$$

Next, the gantry acceleration required to cover the control point spacing, $\Delta \theta_{e}$, in time $t_{\mathrm{del},e}$ is calculated as
(8)$$\alpha_{e}=\frac{2(\Delta \theta_{e}-\left(\omega_{0,e}\times t_{\mathrm{del},e}\right))}{t_{\mathrm{del},e}^{2}},\;e\in E$$
under the constraint that $|\alpha_{e}|$ cannot exceed $\alpha_{\max}$, where $\omega_{0,e}$ is the initial gantry angular velocity at the start of energy layer *e*. The gantry angular velocity can then be calculated for each spot *i* using
(9)$$\omega_{i}=\sqrt{\omega_{i-1}^{2}+2\alpha_{e}(\theta_{i}-\theta_{i-1}}),\;i\in S_{e}.$$
under the constraint that $\omega_{i}$ cannot exceed $\omega_{\max}$ and boundary condition $\omega_{1}$ = 0°s^−1^. Therefore, the overall corrected timestamp of each spot can be calculated as
(10)$$t_{\mathrm{cor},i}=\frac{-\omega_{i-1}+\sqrt{\omega_{i-1}^{2}+2\alpha_{e}\left(\theta_{i}-\theta_{i-1}\right)}}{\alpha_{e}},\;i\in S_{e}.$$

This allows for the angular location of each spot during the continuous delivery of a sawtooth PAT plan to be calculated as $\theta_{i}=\theta \left(t_{\mathrm{cor},i}\right)$. Once delivery of a tooth is complete, the gantry rotates to the angle corresponding to the start of the next tooth. Here, it is assumed the gantry accelerates at $\alpha =\alpha_{\max}/2$ across half the angular distance between the end of the previous tooth and start of the next tooth, $\Delta \theta_{f}$, starting with an initial angular velocity equal to that of the end of the previous tooth, $\omega_{\mathrm{end},f}$. Then, the gantry decelerates over the remaining distance such that it will be stationary at the start of the next tooth. This process occurs in a time
(11)$$t_{\mathrm{dead},f}=t_{\mathrm{acc},f}+t_{\mathrm{dec},f},\;f\in F,$$
where
(12)$$t_{\mathrm{acc},f}=\frac{-\omega_{\mathrm{end},f}+\sqrt{\omega_{\mathrm{end},f}^{2}+\frac{\alpha_{\max}\;\Delta \theta_{f}}{2}}}{\alpha_{\max}},\;f\in F,$$
(13)$$t_{\mathrm{dec},f}=\frac{\Delta \theta_{f}}{\sqrt{\omega_{\mathrm{end},f}^{2}+\frac{\alpha_{\max}\;\Delta \theta_{f}}{2}}},\;f\in F.$$

Here, $t_{\mathrm{dead},f}$ corresponds to the time where the gantry is rotating to the angle corresponding to the start of the next tooth but no beam is being delivered and is set to 0 s if it is lower than $t_{\mathrm{switch},f}$. The overall continuous delivery time of a PAT plan can then be approximated using Equation (Equation 6).

### 2.2. Datasets and Clinical Cases

The delivery of each IMPT and Sawtooth PAT plan developed in a recent study [30] was emulated separately. Specifically, this included IMPT, single arc and dual arc PAT plans for an abdominal phantom (A1) [34] and two cases of ependymoma (E), oropharyngeal (O) and chondrosarcoma (B) cancer. Here, the single arc PAT delivery utilises a single clockwise motion of the gantry during delivery, while dual arc requires a single clockwise rotation followed by a counter-clockwise rotation across the same angular sector. Details of the field arrangements and parameters underpinning these plans are provided in Table A1 in the Appendix B.

While the static IMPT plans were emulated under the clinical standard of step-and-shoot delivery, all PAT plans were emulated using both step-and-shoot and continuous delivery. For each emulated PAT plan under continuous delivery, a separate copy was made where the delivery angle and spot location relative to the isocentre was randomly perturbed, according to Gaussian distributions with mean of 0 mm/° and standard deviation of 0.33 mm/° to evaluate the effect of spot positional/angular shifts that would not be prevented under the current system QC protocol at The Christie NHS Foundation Trust’s PBT facility. During each emulation, the total ELST, spot scanning time, spot delivery time and dead-time were recorded and used to calculate the overall delivery time of each plan in accordance with Equation (Equation 6). As the gantry rotates during delivery of PAT plans under the continuous delivery regime, spots are delivered at a different angle than planned. Therefore, the dose distribution resulting from the emulation of each PAT plan under continuous delivery was calculated, and the relevant dose metrics were recorded and compared to the equivalent step-and-shoot plan. For each anatomical case, a comparison of the resulting dose distributions in the relevant ROIs was also made using matRad’s in-built gamma analysis function using global dose-difference/distance-to-agreement parameters of 1%/1 mm and 3%/3mm and a masking threshold of 1% prescription dose.

## 3. Results

Table 1 shows a breakdown of the time spent on the individual components of plan delivery for the static IMPT, and single and dual PAT plans under the step-and-shoot and continuous delivery regimes for example cases of each anatomical treatment site. The equivalent information for the remaining clinical cases is provided in the Appendix A. Overall, the continuously delivered PAT required a longer delivery time than IMPT, with differences ranging from 29.1–294.6 s and 36.8–442.5 s for single and dual arc delivery, respectively. This difference further increased if step-and-shoot delivery was used, with differences ranging from 121.4–478.3 s to 212.7–831.8 s for single and dual arcs, respectively.

Table 2 outlines the magnitude of the angular and positional shifts each spot incurs during the emulation of continuous PAT delivery for each of the example cases. The angular spot shifts were consistently less than the planned control point spacing, such that each spot was able to be delivered before the gantry reached the location of the next energy layer switch. Further detail on the individual spot shifts and gantry motion during delivery of the E1 PAT plans is provided in the Appendix A. Across all relevant ROIs, a 100% pass-rate between the emulated and planned dose distributions was consistently found from gamma analysis under the dose-difference and distance-to-agreement criteria of 3% and 3 mm, respectively. However, under the tighter criteria of 1% and 1 mm, the pass rate was frequently reduced. This was more prevalent for dual arc delivery due to the increased control point spacing, with a mean pass rate of 94.2% relative to 98.3% for single arc delivery, as exemplified in Figure 1 for the E1 case.

For E1, the clinical dose metrics across all PAT plans under continuous delivery were found to be within 1% of the step-and-shoot plan. As such, all emulated PAT plans were able to meet the corresponding clinical dose requirements for this treatment site. Further details from the dosimetric comparison and gamma analysis of the step-and-shoot and continuously delivered PAT dose distributions across each anatomical case is provided in the Appendix A.

## 4. Discussion

All sawtooth PAT cases included in this study required a longer overall treatment delivery time relative to the corresponding static IMPT plan. This was a result of two factors. Firstly, the developed sawtooth PAT plans consistently utilised a higher number of energy layers than static IMPT such that a higher overall energy layer switching time was required. A previous study found that reducing the number of energy layers (teeth) in the PAT plans resulted in a degradation of plan robustness, potentially to below clinical standards [30]. The fact that continuous single and dual arc PAT plans consistently had more upward energy layer switches with an average contribution to the overall energy layer switching time of ~60% relative to 50% for IMPT means any technological advancements that reduce the current upward ELST of 30 s closer to that of downward ELSTs would reduce the disparity between IMPT and sawtooth PAT delivery times at all facilities with this limitation. However, the clinical deployment of PAT remains the only motivation for this development. Advancements that enable the further reduction of downward ELSTs would also bring sawtooth PAT delivery more in line with the current static IMPT delivery times, albeit to a lesser effect.

Secondly, the delivery of all PAT plans developed on clinical cases was more limited by gantry motion than static IMPT. Specifically, on average, the dead-time for continuous single and dual PAT delivery was 1.28 and 1.40 min longer than IMPT, respectively. This represents an 11% increase in the proportion of the total delivery time where the delivery is being limited by gantry motion. Despite PAT delivery requiring no insertion/retraction of the nozzle, the maximum gantry angular acceleration frequently hinders the delivery of the plan since the gantry is unable to reach the angle required to deliver the next energy layer in the time required to deliver the spots of the current energy layer and switch energy. As such, the gantry rarely reaches its maximum angular velocity. However, in agreement with the previous studies [5], the delivery of sawtooth PAT under a continuous delivery regime has the potential to significantly reduce this dead-time relative to step-and-shoot PAT since the gantry is able to reach higher angular velocities during delivery. Technological improvements that enable a higher angular gantry acceleration and velocity whilst maintaining the current spot delivery accuracy would, therefore, also reduce the disparity between IMPT and sawtooth PAT delivery times. However, any such implementation should ensure that patient comfort and safety are maintained. Unlike the methods developed by Qian et al. [24] and Liu et al. [35], the emulator makes no attempt to smooth out the rotation velocity of the gantry during delivery. This method of delivery may increase the delivery burden on the gantry and increase the likelihood of overshooting from the correct angular position during delivery.

Since all of the existing studies into the delivery time of PAT are based on a synchro-cyclotron IBA ProteusONE^®^ system, the technological differences compared to a cyclotron-based Varian Probeam^®^ system will likely impact the overall delivery time of PAT plans. While gantry motion [27], spot delivery times and scanning rates [21] are similar, the fact both systems use alternate methods of energy selection causes significant differences in ELST. Specifically, at the Christie NHS Foundation Trust PBT facility, we observed a linear relationship between the magnitude of a downward energy switch and its corresponding switching time to the order of 0.5–3 s, whereas an initial energy dependent step-like response was observed at a given threshold energy difference on the IBA ProteusONE^®^ system [21], beyond which downward ELSTs increased to ∼ 3.5 s. However, upward energy layer switches on the Christie NHS Foundation Trust’s Varian Probeam^®^ system require manual intervention, taking 30 s regardless of the initial or final energy, significantly higher than the 4.5–6.5 s observed on an IBA ProteusONE^®^ system [21]. Therefore, in order to improve the clinical feasibility of delivering a PAT plan on a current Varian ProBeam^®^ system, the number of upward energy switches within the PAT plan must be reduced as much as clinically possible. For this reason, since SPArc plans commonly contain 9–26 upward energy layer switches [5,6,14,36], it is likely the delivery times on a Varian ProBeam^®^ system would be significantly higher than the developed sawtooth PAT plans.

The inclusion of random angular and spatial permutations on the location of each spot increased these differences relative to the step-and-shoot PAT plan by 0.03 ± 0.04° and 0.45 ± 0.16 mm, causing no clinically significant differences relative to the continuously delivered sawtooth PAT dose distributions. This shows that the precision of current Varian ProBeam^®^ systems is appropriate for the continuous delivery of sawtooth PAT plans. Since this study aimed to evaluate the dosimetric consequences of delivering sawtooth PAT plans, each plan was not re-optimised post emulation. Once the spot positional shifts during delivery have been calculated, we expect that re-optimisation under the same planning objectives would correct for the observed dose distribution changes. This process could start from the existing spot weights such that the optimisation time would be significantly less than during planning.

It is also important to note that all the plans involved in this study required a couch position of 0°. As such, the emulator does not evaluate the time required to reposition the couch between fields/teeth. At the Christie NHS Foundation Trust’s PBT facility, the clinical standard when switching fields during static IMPT delivery involves validating the couch movement using 2D kV imaging. During this time, as this is a multi-room centre, it is common for the beam to switch between rooms. This process can significantly increase the overall in-room treatment time and be highly dependent on whether the facility is currently operating at capacity. As sawtooth PAT is unlikely to require couch moves, in order to reduce the overall delivery time and prevent the interruption of delivery, the beam should not be relinquished to other treatment rooms after delivery of each tooth. Since the emulator does not account for these factors, the results presented represent an evaluation of a single room centre with no gantry/couch validations during treatment.

## 5. Conclusions

We have developed an IMPT and PAT emulator in order to model the delivery of static IMPT, as well as single and dual arc sawtooth PAT plans under a step-and-shoot and continuous delivery regime, on a cyclotron-based PBT system. This emulator was applied to the previously developed static IMPT and single and dual arc sawtooth PAT plans for an abdominal CT phantom and multiple cases of brain, base of skull (BoS) and head and neck (H&N) cancers. This enables a comparison between the time required for individual components of treatment for PAT and static IMPT plans, as well as an evaluation of the dosimetric consequences of continuous PAT delivery.

The overall delivery time of all clinical single and dual arc PAT plans was consistently higher than the corresponding static IMPT plan. This was primarily due to two factors: Firstly, most PAT plans had more teeth than fields in the corresponding IMPT plan, resulting in a larger overall energy layer switching time (ELST). This effect was most prominent for targets with complex geometries that were in close proximity to OARs (BoS and H&N) and is expected to apply to all treatment sites of this nature. Secondly, PAT plans were more limited by the ability of the gantry rotation to keep up with delivery, increasing the overall dead-time during treatment. The latter was significantly increased under a step-and-shoot delivery regime. This is likely to be a general property of PAT across all treatment sites on the current clinical systems. Therefore, since continuous PAT delivery showed no clinically significant dosimetric differences relative to the planned (step-and-shoot) dose distribution, this study suggests that continuous delivery is likely to be the most suitable method of PAT delivery. Future improvements in accelerator and gantry technology that reduce upward ELSTs and enable a more rapid gantry movement while maintaining delivery accuracy may provide evidence towards using alternate methods of PAT delivery optimisation.

## Figures and Tables

**Figure 1 cancers-16-03315-f001:**
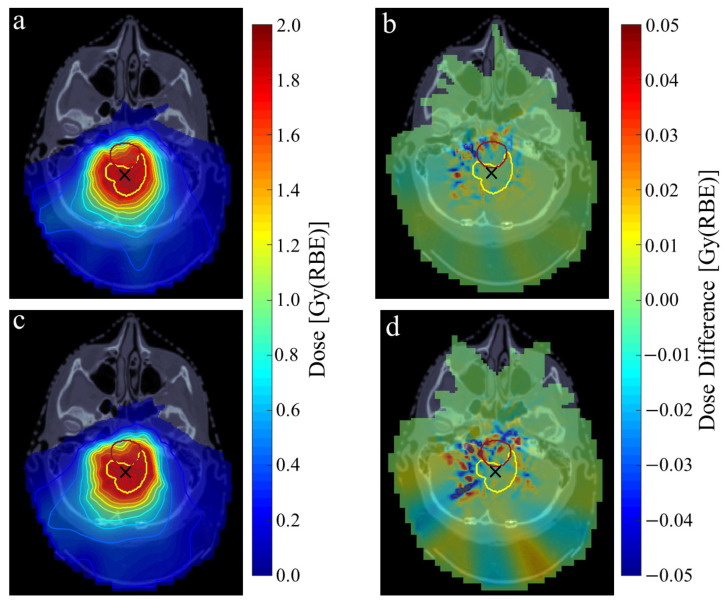
Single-fraction dose distribution of the ependymoma E1 case under continuous single (**a**) and dual (**c**) arc PAT delivery. Dose differences of each plan relative to the planned dose distribution of step-and-shoot PAT are shown in the right-hand column (**b**,**d**). CTV and brainstem contours shown in yellow and red, respectively. Isocentre marked using black cross. Isodose lines shown in 10% and 5% intervals for 10–90% and 90–105% single-fraction dose, respectively.

**Table 1 cancers-16-03315-t001:** Breakdown of the emulated time spent on each part of delivery for each static IMPT plan and single and dual arc PAT plan under the step-and-shoot and continuous delivery regimes for each example dataset. Errors correspond to one standard deviation deriving from variations in the experimental measurements of ELST, and MU delivery rates are shown. Abbreviations: EL = energy layer, SAS = step-and-shoot, CON = continuous.

Dataset	Delivery Method		EL Switching [s]	Spot Scanning [s]	Spot Delivery [s]	Dead-Time [s]	Total [s]
A1	Static IMPT		106.5 ± 8.1	2.1	16.0 ± 1.4	69.9	194.5 ± 9.5
	PAT single arc	SAS	145.5 ± 9.6	2.1	15.5 ± 1.4	152.7	315.9 ± 11.0
		CON				60.4	223.6 ± 11.0
	PAT dual arc	SAS	148.1 ± 10.1	2.0	15.3 ± 1.4	241.7	407.2 ± 11.4
		CON				65.9	231.3 ± 11.4
E1	Static IMPT		105.3 ± 7.9	1.9	13.7 ± 1.2	0.0	120.9 ± 9.1
	PAT single arc	SAS	221.6 ± 13.0	2.2	9.9 ± 0.9	278.8	513.5 ± 13.5
		CON				91.2	325.9 ± 13.5
	PAT dual arc	SAS	228.2 ± 13.5	2.1	10.6 ± 0.95	432.8	673.7 ± 14.5
		CON				101.4	341.9 ± 14.5
B1	Static IMPT		184.6 ± 16.4	4.9	26.4 ± 2.4	0.0	215.9 ± 18.8
	PAT single arc	SAS	312.7 ± 23.0	3.3	22.6 ± 2.0	354.6	694.2 ± 25.0
		CON				149.5	489.1 ± 25.0
	PAT dual arc	SAS	427.6 ± 27.3	3.8	20.3 ± 1.8	595.4	1047.2 ± 29.1
		CON				206.7	658.4 ± 29.1
O1	Static IMPT		252.1 ± 22.9	12.9	148.4 ± 13.3	64.1	447.5 ± 36.2
	PAT single arc	SAS	389.6 ± 31.1	14.3	152.8 ± 13.7	369.3	924.9 ± 4.8
		CON				186.5	742.1 ± 44.8
	PAT dual arc	SAS	264.0 ± 19.8	9.6	146.3 ± 13.1	552.6	972.6 ± 32.9
		CON				118.4	538.3 ± 32.9

**Table 2 cancers-16-03315-t002:** Summary of the mean and standard deviation in the angular and positional spot shifts resulting from continuous PAT delivery for each example case. Mean gamma pass rates across all relevant targets and OARs using a dose-difference and distance-to-agreement criteria of 1% and 1 mm are shown. Abbreviations: CP = control point.

Dataset	Delivery Method	CP Spacing [°]	Spot Shifts		γ Pass Rate
			[°]	[mm]	
A1	Single Arc	0.8	0.33 ± 0.28	0.82 ± 2.52	99.3 ± 0.15
	Dual Arc	1.6	0.65 ± 0.50	1.07 ± 4.03	97.4 ± 0.32
E1	Single Arc	1.1	0.42 ± 0.33	1.04 ± 1.75	99.3 ± 1.12
	Dual Arc	2.2	0.86 ± 0.69	1.55 ± 2.03	96.8 ± 5.02
B1	Single Arc	0.7	0.25 ± 0.28	1.40 ± 2.80	95.8 ± 4.80
	Dual Arc	1.0	0.29 ± 0.31	1.50 ± 2.84	94.9 ± 6.68
O1	Single Arc	0.6	0.26 ± 0.19	0.96 ± 2.05	98.8 ± 1.38
	Dual Arc	1.6	0.81 ± 0.59	1.78 ± 2.44	87.6 ± 10.22

## Data Availability

Data supporting this study are included within the article. The derived data supporting the findings of this study are available upon reasonable request from the corresponding author, Samuel Burford-Eyre.

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
