# Peer review of "Emulating the Delivery of Sawtooth Proton Arc Therapy Plans on a Cyclotron-Based Proton Beam Therapy System"

_cancers, 2024, doi:10.3390/cancers16193315_

Round 1
Reviewer 1 Report
Comments and Suggestions for Authors
The authors present a manuscript describing an emulator for evaluating beam-on time for various proton therapy delivery techniques with the Varian Probeam system, most notably proton arc therapy (PAT). The emulator is certainly a useful tool to evaluate and optimize PAT delivery techniques. However, a particular challenge with the review of this manuscript was that the work was focused on the authors’ sawtooth PAT technique, and the description of this technique is within a manuscript not yet published.
The authors also discuss an experimental delivery of the sawtooth PAT technique. While impressive, it is not clear how this experimental delivery adds to the work given the significant differences between the fixed experimental beamline on which it was delivered and clinical gantry.
While the manuscript is very thorough and well written, I believe it could be significantly shortened without compromising the quality and contribution to the field.
Specific comments:
1. Page 2, line 77. It was not clear how the content of this paragraph was relevant to the current work. I would suggest removing.
2. Page 4, line 143. Is the static IMPT referred to here applicable to both fixed-field IMPT and step-and-shoot PAT? If this is specifically referring to fixed-field IMPT please use consistent wording.
3. Section 2.2. It is not clear what this experimental delivery adds to the study. The rotation stage does not appear to provide a realistic model of the gantry acceleration and velocity. I would suggest removing unless the authors provide further justification for inclusion.
4. Section 3. Was the emulator validated against actual delivery times for the fixed-field IMPT plans? This needs to be included in the results.
5. Page 8, line 299. The description for Figure 3 does not match the caption of Figure 3. The body suggests b) and d) represent differences between planned and emulated doses. The caption suggests b) and d) represent differences between continuous PAT and step and shoot PAT. The same applies to Figure 4 and Figure 5.
6. Section 3.2. Given the extreme differences in results between experimental delivery and emulation, the relevance of this data was difficult to interpret. Again, suggest the experimental delivery be removed if it does not add anything meaningful.
7. Page 15, line 521. Here it states that emulated beam-on time was within 2s of experimental delivery. This does not appear to be in keeping with the results.
Author Response
The authors present a manuscript describing an emulator for evaluating beam-on time for various proton therapy delivery techniques with the Varian Probeam system, most notably proton arc therapy (PAT). The emulator is certainly a useful tool to evaluate and optimize PAT delivery techniques. However, a particular challenge with the review of this manuscript was that the work was focused on the authors’ sawtooth PAT technique, and the description of this technique is within a manuscript not yet published.
>> While a manuscript outlining the details of the Sawtooth PAT planning algorithm is currently under review by the same journal, we recognize that including a brief description on the basic principles of the Sawtooth PAT planning algorithm in the current version of the manuscript would improve the manuscript’s transparency and accessibility to the reader. We have therefore added a brief description of the main steps included within the Sawtooth PAT planning algorithm into the manuscript on page 3, lines 105 - 112.
The authors also discuss an experimental delivery of the sawtooth PAT technique. While impressive, it is not clear how this experimental delivery adds to the work given the significant differences between the fixed experimental beamline on which it was delivered and clinical gantry.
>> The aim of analyzing the delivery of a PAT plan on the fixed experimental beamline was to validate the emulator’s spot delivery time model and the experimental methodology that was used to measure the MU delivery rates included by the model, as outlined in the supplementary material. As the fixed experimental beamline is linked to the same cyclotron as the clinical gantries and utilizes the same ProBeam® nozzle, the same methods can be used to develop such models, albeit with differing values due to differences in scanning slew-rates and beam transmission. As we consistently measured the experimental beam delivery time to be within 2s of the emulator, we are confident that the beam delivery of the RR emulator are realistic and therefore that the underlying experimental methodology is sound. However, we recognize that this observation alone is not sufficient to justify the inclusion of the experimental delivery of Sawtooth PAT plans within this manuscript. The additional points outlined by this reviewer on the suitability of using the rotation stage to model clinical gantry motion and the length of the manuscript also reflect this. Therefore, we have removed all references to the experimental delivery of Sawtooth PAT plans from this manuscript and relabeled the sections accordingly.
While the manuscript is very thorough and well written, I believe it could be significantly shortened without compromising the quality and contribution to the field.
>> We thank the reviewer for their positive comments regarding our manuscript. We have removed all references to the experimental delivery of Sawtooth PAT plans from this manuscript. We believe this change shortens the manuscript without compromising quality.
Specific comments:
1. Page 2, line 77. It was not clear how the content of this paragraph was relevant to the current work. I would suggest removing.
>> We agree that the discussion on PAT QC and QA tests provided in the latter stages of this paragraph may be outside the direct scope of the main aim of the manuscript. Therefore, we have removed the last two sentences of this paragraph and added the first three sentences of the following paragraph to improve readability.
2. Page 4, line 143. Is the static IMPT referred to here applicable to both fixed-field IMPT and step-and-shoot PAT? If this is specifically referring to fixed-field IMPT please use consistent wording.
>> We recognise that the use of ‘fixed-field’ and ‘static’ IMPT terms were being used synonymously throughout the manuscript and may cause confusion to the reader. The case the reviewer refers to was only applicable to fixed-field IMPT. Therefore we have modified to manuscript to only use the term ‘static’ when referring to conventional IMPT delivery to improve the consistency and readability of the manuscript.
3. Section 2.2. It is not clear what this experimental delivery adds to the study. The rotation stage does not appear to provide a realistic model of the gantry acceleration and velocity. I would suggest removing unless the authors provide further justification for inclusion.
>> We outline the motivation for the experimental delivery of the Sawtooth PAT plans in our answer to the second general comment from the reviewer. The intended use of the rotation stage was to monitor the gantry position and motion during the experimental delivery. However, we recognise that the rotation stage does not represent an accurate model of gantry motion due to its pseudo-instantaneous acceleration. This is highlighted by the significant difference in the mean rotation velocities between the stage during delivery and the emulated gantry motion. We have removed all references to the experimental delivery of Sawtooth PAT plans from this manuscript.
4. Section 3. Was the emulator validated against actual delivery times for the fixed-field IMPT plans? This needs to be included in the results.
>> Unfortunately the overall delivery times for the fixed-field IMPT plans were not compared to the true clinical delivery times for the cases involved in this manuscript that received treatment at The Christie NHS Foundation Trust. The main reason for this is that it is common practice at our facility to switch the beam between rooms while rotating the gantry to the angle associated with the next field ahead of delivery. The waiting time required before beam is restored to the gantry for delivery is dependent on multiple factors such as whether the facility is operating at capacity and the types of cases being treated in the other gantries. Therefore, we believe such a comparison would not provide any reliable information towards validating the emulator’s predicted delivery times. We recognise that incorporating such factors within the emulator would improve its suitability for use at multi-gantry facilities. However, this wasn’t the primary aim of the presented work and we explicitly state that the emulator delivery times only represent that of a single room centre on page 13, line 442.
5. Page 8, line 299. The description for Figure 3 does not match the caption of Figure 3. The body suggests b) and d) represent differences between planned and emulated doses. The caption suggests b) and d) represent differences between continuous PAT and step and shoot PAT. The same applies to Figure 4 and Figure 5.
>> The captions are correct as the planned doses result from the delivery of step-and-shoot PAT. We recognise that the current wording is confusing, especially due to the absence of significant details of the Sawtooth PAT planning algorithm in this manuscript. Therefore, we have updated the body to explicitly state this on page 7, line 253; page 9, line 297; and page 11, line 326.
6. Section 3.2. Given the extreme differences in results between experimental delivery and emulation, the relevance of this data was difficult to interpret. Again, suggest the experimental delivery be removed if it does not add anything meaningful.
>> As discussed under review 1 question 3, we agree with the reviewer that the methodology and analysis of the experimental delivery of Sawtooth PAT plans does not add sufficient additional information on the accuracy of the emulator to warrant inclusion in this manuscript. Therefore all references to the experimental delivery have been removed.
7. Page 15, line 521. Here it states that emulated beam-on time was within 2s of experimental delivery. This does not appear to be in keeping with the results.
>> This is true, the results were that the measured beam on time was considerably higher than the emulated beam-on time of 2 s. We later discuss the cause of this in the Discussion section on page 13, lines 413-424. The majority of the observed beam on time was due to the time required to reduce the beam current down to 0 nA before energy switching takes place (time taken to initiate the vertical deflector). However, as discussed under review 1 question 3, all aspects of the experimental delivery of Sawtooth PAT plans have been removed from the manuscript.
Reviewer 2 Report
Comments and Suggestions for Authors
I thank the editors for the opportunity to review this manuscript, and the authors for their work in preparing it. The study focusses on the determination and evaluation of the treatment delivery time of a variety of IMPT as well as “sawtooth” proton arc therapy treatment plans, and finds considerably higher treatment delivery times for step-and-shoot as well as continuous delivery PAT compared to IMPT.
The manuscript is very interesting and well-written and generally clear throughout, and only a few comments with regards to the methodology and/or presentation come to mind:
Major comments:
- The manuscript and its conclusions are specific to the “sawtooth” PAT treatment planning strategy applied, but the current version of the manuscript does not elaborate upon this planning approach, and the manuscript describing the approach (reference #29) is still under review. I therefore think it may be necessary that the manuscript be expanded with an abridged explanation of the sawtooth treatment planning strategy so that the reader can gain a better understanding of it and the resulting implications for the treatment delivery time.
- One main explanation for the higher delivery time of PAT compared to IMPT was the higher number of energy layers associated with the former. The authors indicate that the other manuscript currently under review shows that a reduction in the number of energy layers degraded plan robustness, but given that this manuscript’s conclusions heavily rely on the number of energy layers and the idea that a reduction in the number of energy layers may degrade plan robustness below acceptable levels, I think that expanding the manuscript with additional treatment plans with a reduced number of energy layers to further investigate the trade-off between delivery time and plan robustness to determine how far the number of energy layers can be reduced before an unacceptable degradation in plan quality is warranted.
- Lines 480-491: Given that the potential differences in clinical IMPT and PAT workflows mentioned here may result in relevant temporal differences (in comparison to the determined treatment times), I think that a brief sensitivity analysis with respect to realistic values for e.g. delays resulting from beam switching between rooms and the resulting changes in total treatment times would strengthen the manuscript considerably.
Minor comments:
- Line 23: Upon first reading, it may not immediately be clear to the reader why the spot position accuracy is specified in degrees, so a brief expansion of this sentence may be helpful.
- Line 51: The abbreviation ELST is used before it is introduced.
- Lines 369-376: Without part of the explanation later provided in the discussion, the magnitude of the differences mentioned here may appear confusing, so I think it may be helpful to already provide a very brief explanation in this paragraph.
- Line 521: Specifically, after correcting the initial delivery model as described in lines 421-424?
- Table A1: To better understand their implications with respect to the treatment delivery time, I also think it would be worthwhile to expand the appendix’ Table A1 with additional information such as the number of energy layers, the number of pencil-beam spots etc.
- Generally: The manuscript’s novelty lies in the investigated delivery system, so I think that emphasizing that the findings are specific to this delivery model in the abstract’s and the manuscript’s conclusion may be warranted.
Author Response
Major comments:
- The manuscript and its conclusions are specific to the “sawtooth” PAT treatment planning strategy applied, but the current version of the manuscript does not elaborate upon this planning approach, and the manuscript describing the approach (reference #29) is still under review. I therefore think it may be necessary that the manuscript be expanded with an abridged explanation of the sawtooth treatment planning strategy so that the reader can gain a better understanding of it and the resulting implications for the treatment delivery time.
>> While a manuscript outlining the details of the Sawtooth PAT planning algorithm is currently under review by the same journal, we recognize that including a brief description on the basic principles of the Sawtooth PAT planning algorithm in the current version of the manuscript would improve the manuscript’s transparency and accessibility to the reader. We have therefore added a brief description of the main steps included within the Sawtooth PAT planning algorithm into the manuscript on page 3, lines 105 - 112.
- One main explanation for the higher delivery time of PAT compared to IMPT was the higher number of energy layers associated with the former. The authors indicate that the other manuscript currently under review shows that a reduction in the number of energy layers degraded plan robustness, but given that this manuscript’s conclusions heavily rely on the number of energy layers and the idea that a reduction in the number of energy layers may degrade plan robustness below acceptable levels, I think that expanding the manuscript with additional treatment plans with a reduced number of energy layers to further investigate the trade-off between delivery time and plan robustness to determine how far the number of energy layers can be reduced before an unacceptable degradation in plan quality is warranted
>> The balance of dosimetric plan robustness and delivery time (due to energy layer switching) is currently the primary focus of PAT research. The reviewer’s recommendation of reducing the number of energy layers until an unacceptable degradation in robustness is observed is directly evaluated in our additional manuscript that is currently under review by the same journal. The Sawtooth PAT plans included in this manuscript contain a number of upward energy switches such that further increasing the number of upward energy switches does not improve the dosimetric plan quality and reducing the number of upward energy switches leads to a significant degradation.
- Lines 480-491: Given that the potential differences in clinical IMPT and PAT workflows mentioned here may result in relevant temporal differences (in comparison to the determined treatment times), I think that a brief sensitivity analysis with respect to realistic values for e.g. delays resulting from beam switching between rooms and the resulting changes in total treatment times would strengthen the manuscript considerably.
>> The increase in clinical IMPT delivery times due to the beam switching between rooms is dependent on multiple factors such as whether the facility is operating at capacity and the types of cases being treated in the other gantries. These factors are highly variable and centre-dependent such that ‘realistic values’ can range from 30 s up to the delivery of an entire treatment fraction (20 mins). Therefore, a large scale departmental review of treatment delivery times would be required across all patients in order to determine such ‘realistic values’. We believe such a comparison would not provide any reliable information towards validating the emulator’s predicted delivery times. We recognise that incorporating such factors within the emulator would improve its suitability for use at multi-gantry facilities. However, this wasn’t the primary aim of the presented work and we explicitly state that the emulator delivery times only represent that of a single room centre on page 13, line 442.
Minor Comments:
- Line 23: Upon first reading, it may not immediately be clear to the reader why the spot position accuracy is specified in degrees, so a brief expansion of this sentence may be helpful.
>> As discussed under review 1 question 3, all information detailing the methodology, results and discussion of the experimental delivery of Sawtooth PAT plans have been removed from the manuscript. The main reasons for this were to shorten the manuscript, and that this work did not provide significant benefit to since the methodology didn’t allow the robust evaluation of the emulator’s delivery time predictions.
- Line 51: The abbreviation ELST is used before it is introduced.
>> We have defined the ELST abbreviation at the location outlined.
- Lines 369-376: Without part of the explanation later provided in the discussion, the magnitude of the differences mentioned here may appear confusing, so I think it may be helpful to already provide a very brief explanation in this paragraph.
>> As discussed under review 1 question 3, we have removed all references to the experimental delivery of Sawtooth PAT plans from our manuscript.
- Line 521: Specifically, after correcting the initial delivery model as described in lines 421-424?
>> As discussed under review 1 question 3, we have removed all references to the experimental delivery of Sawtooth PAT plans from our manuscript.
- Table A1: To better understand their implications with respect to the treatment delivery time, I also think it would be worthwhile to expand the appendix’ Table A1 with additional information such as the number of energy layers, the number of pencil-beam spots etc.
>> We have added the number of total energy layers and spots corresponding to each plan outlined in Table A1 to provide further background to the reader on the details of each plan when evaluating the results of the manuscript and updated the table caption accordingly.
- Generally: The manuscript’s novelty lies in the investigated delivery system, so I think that emphasizing that the findings are specific to this delivery model in the abstract’s and the manuscript’s conclusion may be warranted.
>> We believe that the approach taken in the work outlined in this manuscript is applicable to any cyclotron-based PBT delivery system. Specifically both the Varian ProBeam® and IBA Proteus Plus® use similar beamline components and gantry motion parameters. Therefore, while the specific measurements underpinning the emulator were taken on a ProBeam® system, it is feasible that the underlying methodology be relevant to facilities with other similar systems.
Round 2
Reviewer 2 Report
Comments and Suggestions for Authors
Some of my prior comments were addressed, but after the recent changes, some parts of the manuscript now contradict other parts. Most importantly, the manuscript's title, abstract, and introduction still refer to an experimental validation of the emulator, although this aspect has now been removed from the rest of the manuscript.
With the removal of the emulator's experimental validation, I believe that an expansion of the manuscript in a different direction is necessary. Since the potentially longer treatment delivery times often go hand-in-hand with dosimetric advantages, I am of the opinion that both should be reported together, and that it would be most natural to merge both manuscripts to provide a more complete picture. Alternatively, this manuscript could be expanded to include modelled treatment delivery times as at least one treatment plan parameter (e.g. number of energy layers) is modified. Since, as I understand it, such treatment plans were created for the other manuscript currently under review, applying the emulator to a greater variety of treatment plans and thereby giving the reader a better understanding of the treatment delivery time's dependence on at least one treatment plan parameter seems very feasible.
Author Response
Some of my prior comments were addressed, but after the recent changes, some parts of the manuscript now contradict other parts. Most importantly, the manuscript's title, abstract, and introduction still refer to an experimental validation of the emulator, although this aspect has now been removed from the rest of the manuscript.
With the removal of the emulator's experimental validation, I believe that an expansion of the manuscript in a different direction is necessary. Since the potentially longer treatment delivery times often go hand-in-hand with dosimetric advantages, I am of the opinion that both should be reported together, and that it would be most natural to merge both manuscripts to provide a more complete picture. Alternatively, this manuscript could be expanded to include modelled treatment delivery times as at least one treatment plan parameter (e.g. number of energy layers) is modified. Since, as I understand it, such treatment plans were created for the other manuscript currently under review, applying the emulator to a greater variety of treatment plans and thereby giving the reader a better understanding of the treatment delivery time's dependence on at least one treatment plan parameter seems very feasible.
>> We have removed all references to the experimental delivery of Sawtooth PAT from the title, abstract and introduction of this manuscript.
We recognise the reviewer’s opinion that the Sawtooth PAT delivery time analysis presented in this manuscript may benefit from being combined with our other manuscript presenting the Sawtooth PAT planning algorithm and making dosimetric comparisons relative to fixed-field IMPT. However, we believe that this combination would result in an excessively long manuscript, potentially confusing the reader as to the overall aim of the work as well as deviating from the original intended scope of both manuscripts. Furthermore, since our other manuscript remains out for review, we are cautious that the proposed change may interfere with the review process for this manuscript.
While an expansion to the manuscript that investigates the dependence of the overall Sawtooth PAT plan delivery times relative to treatment plan parameters (such as the number of energy layers) is feasible, this topic has already been covered in existing publications for other PAT planning algorithms [1-4]. For example, Zhang et al (2022) showed that the treatment delivery time was only mildly related to the number of energy layers contained within a plan. More important was the number of upward energy layer switches that were required to deliver the plan, since the upward energy switching time is considerably longer than downward energy switching times on current systems. Therefore, we believe that such an expansion would not provide any further benefit to the wider scientific community.
[1] Liu, G., Li, X., Zhao, L., Zheng, W., Qin, A., Zhang, S., Stevens, C., Yan, D., Kabolizadeh, P. and Ding, X., 2020. A novel energy sequence optimization algorithm for efficient spot-scanning proton arc (SPArc) treatment delivery. Acta Oncologica, 59(10), pp.1178-1185.
[2] Wuyckens, S., Zhao, L., Saint-Guillain, M., Janssens, G., Sterpin, E., Souris, K., Ding, X. and Lee, J.A., 2022. Bi-criteria Pareto optimization to balance irradiation time and dosimetric objectives in proton arc therapy. Physics in Medicine & Biology, 67(24), p.245017.
[3] Zhang, G., Long, Y., Lin, Y., Chen, R.C. and Gao, H., 2023. A treatment plan optimization method with direct minimization of number of energy jumps for proton arc therapy. Physics in Medicine & Biology, 68(8), p.085001.
[4] Gu, W., Ruan, D., Lyu, Q., Zou, W., Dong, L. and Sheng, K., 2020. A novel energy layer optimization framework for spot‐scanning proton arc therapy. Medical physics, 47(5), pp.2072-2084.
Round 3
Reviewer 2 Report
Comments and Suggestions for Authors
I believe that, in its current form, the manuscript would be more suitable for publication as short communications
or a report. This is because, with respect to scope, previous publications which focused
on proton arc therapy delivery time assessments for treatment plans generated with different proton arc therapy treatment planning algorithms
were already broader in focus, including the consideration of dosimetric aspects, a higher number of patients studied, variation of treatment machine
parameters, and/or experimental validation of the treatment delivery time model (see e.g. https://www.sciencedirect.com/science/article/abs/pii/S0360301616331649,
https://www.redjournal.org/article/S0360-3016(21)02302-6/fulltext, https://pubmed.ncbi.nlm.nih.gov/37774715/,
https://aapm.onlinelibrary.wiley.com/doi/abs/10.1002/mp.16879). The novelty in such an expansion would lie in the treatment planning algorithm
and treatment delivery system considered, analogously to the novelty of the current draft of the manuscript compared to prior works in the field.
Author Response
I believe that, in its current form, the manuscript would be more suitable for publication as short communications or a report. This is because, with respect to scope, previous publications which focused on proton arc therapy delivery time assessments for treatment plans generated with different proton arc therapy treatment planning algorithms were already broader in focus, including the consideration of dosimetric aspects, a higher number of patients studied, variation of treatment machine parameters, and/or experimental validation of the treatment delivery time model (see e.g. https://www.sciencedirect.com/science/article/abs/pii/S0360301616331649,
https://www.redjournal.org/article/S0360-3016(21)02302-6/fulltext, https://pubmed.ncbi.nlm.nih.gov/37774715/,
https://aapm.onlinelibrary.wiley.com/doi/abs/10.1002/mp.16879). The novelty in such an expansion would lie in the treatment planning algorithm and treatment delivery system considered, analogously to the novelty of the current draft of the manuscript compared to prior works in the field.
>> We have received a response from the reviewers of our manuscript presenting the details of the Sawtooth PAT planning algorithm. As the reviewers did not make any requests for a delivery time analysis to be included within this manuscript, we believe it is inappropriate for our two manuscripts to be combined in order to meet the change of scope previously requested. However, we have made significant changes to the current manuscript in order to convert it into the correct format for publication as a short report. The specific modifications are as follows:
- Figures 1, 3, 4, 5 and A1 have been moved to the supplementary material.
- The length of Section 3 “Results” has been reduced by moving the detailed discussion of the results from the emulation of the individual A1, E1, B1 and O1 cases to the supplementary material. This was replaced by a more brief presentation of the results across these treatment sites as well as a new table (Table 2) to more efficiently present the dosimetric gamma analysis and spot shift observations.
- Paragraphs 4 (page 12-13, lines 393 – 410) and 6 (page 12, lines 422-430) from Section 4 “Discussion” have been removed from the manuscript to reduce its length and prioritise what we view are the most important points of discussion that result from this work.
- The supplementary materials section on page 10 has been modified accordingly to highlight these changes.